# Capitalising on shared goals for family planning: a concordance assessment of two global initiatives using longitudinal statistical models

Qingfeng Li  ,[1,2] Jose G Rimon,[2] Saifuddin Ahmed[2]

¹International Health, Johns Hopkins Bloomberg School of Public Health, Baltimore, Maryland, USA
²Population, Family and Reproductive Health, Johns Hopkins Bloomberg School of Public Health, Baltimore, Maryland, USA

**Correspondence to**
Dr Qingfeng Li; qli28@jhu.edu

## ABSTRACT

**Objective** Family planning is unique among health interventions in its breadth of health, development and economic benefits. The complexity of formulating effective strategies to promote women's and girls' access to family planning calls for closer coordination of resources and attention from all stakeholders. Our objective was to quantify the concordance of two global initiatives: Family Planning 2020's adding 120 million modern contraceptive users by 2020 (proposed during The London Summit 2012 by Gates Foundation) and satisfying the 75% demand for modern contraceptives by 2030 (proposed by United States Agency for International Development). A demonstration of their concordance, or lack thereof, provides an understanding of the proposed quantitative goals and helps to formulate collective strategies.

**Design and setting** We applied fixed effects longitudinal models to assess the convergence of the two initiatives. The implications of success in one initiative on achieving the other are simulated to illustrate their shared goals. Publicly available data on contraceptive use, unmet need and met need from national surveys are used. Extensive model validations were conducted to check and confirm models' predictive performance.

**Results** Our results show that the 75% demand satisfied initiative will reach 82 million additional modern users by 2020 and 120 million by early 2023. Following FP2020's proposed annual increase of modern contraceptive use, 9 of the 41 commitment-making countries will reach the 75% target by 2020; another 8 countries will do so by 2030. Extending FP2020's proposed contraceptive growth to 2030 implies the achievement of the 75% target in less than half (17) of the 41 commitment-making countries.

**Conclusion** The results from the statistical exercise demonstrate that the two global initiatives move toward the same goal of promoting access to family planning and overall both are ambitious. Closer coordination between major stakeholders in international family planning may stimulate more efficient mobilisation and utilisation of global sources, which is urgently needed to accelerate the progress toward satisfying women's need for family planning.

## INTRODUCTION

Access to family planning is a critical component of reproductive rights and leads to multifaceted benefits for women and their families. It is unique among health interventions in its breadth of health, development and economic benefits, such as reducing maternal and child mortality, empowering women and girls, and enhancing environmental sustainability.[1 2] The Lancet series on family planning in 2012 documented strong evidence of the extensive gains resulting from family planning. Ahmed and colleagues[2] estimated that contraceptive use in 172 countries averted 272040 maternal deaths in 2008, and satisfying unmet need for contraceptive methods could prevent another 104000 deaths per year. Cleland and colleagues[3] made nearly identical estimates using a different methodology. Additionally, Canning and Schultz[4] evaluated the economic consequences of family planning, including increases in female labour force participation and proportion in paid employment.

However, after reaching their global peak following the 1994 International Conference on Population and Development (ICPD) in Cairo, both financial support and political commitments for family planning have plateaued, and even declined in many countries, in the decade prior to 2012.[1 5] Consequently, progress towards providing access to contraception for women and girls in developing countries has been slow. On average,

women in Sub-Saharan Africa continue to have more than five children.[6]

**Strengths and limitations of this study**

► This study is the first systematic comparison of two major global initiatives on family planning.
► The estimations are based on rigorously developed and validated statistical models.
► The findings provide new insights into the shared goals of the two initiatives and have important policy implications.
► Relying on secondary data restricts variable selection for the statistical models.
► The linear growth curve assumed for modern contraceptive prevalence rate in the study may not be accurate for each country.

Compared with other public health interventions, family planning has two unique features that need special attention. First, due to cultural, religious and political reasons, family planning is more controversial than many other public health issues.[1] Even the proponents of family planning disagree with each other over what the primary aims should be. Some emphasise ecological concerns, specifically the effect of fertility declines on population structure, ecosystem and economy. Others emphasise human rights concerns, promoting women's control over their own reproduction.[7]

Second, unlike other public health issues, such as reducing child mortality, the biomedical side of family planning is well-established, with proven methods to space and limit pregnancies. Where the successful implementation of family planning programme is concerned, it has been established that a key element is a political issue of obtaining support from and forming a broad coalition of elite groups.[1 7] This has proven successful in many countries, but remains elusive in some, especially in Sub-Saharan Africa.

The complexity of formulating effective strategies to promote women's and girls' access to family planning calls for closer coordination of resources and attention from all stakeholders. As noted by Kim and Ammann,[8] a clear consensus on targets and priorities is indispensable for all successful public projects in the modern era.

During the past few years, two major family planning initiatives were launched. First, the London Summit on Family Planning in July 2012 was convened by the United Kingdom's Department for International Development (DFID) and the Bill and Melinda Gates Foundation (BMGF). At the Summit, leaders proposed adding 120 million female modern contraceptive users in the world's 69 poorest countries by 2020.[9] The second initiative, led by the United States Agency for International Development (USAID), proposed a target of satisfying 75% of the demand for family planning with modern contraceptives by 2030.[10 11] This indicator of demand satisfied was subsequently adopted in the sustainable development goals.[12] The % demand satisfied is the proportion of women who use modern contraception divided by the total demand for family planning, which is defined by adding the percentage of married or in-union women aged 15–49 years who are using any contraception to the percentage of women with unmet need. Unmet need refers to the proportion of women who want to stop or delay childbearing but are not using any method of contraception. Following Fabic et al,[10] in the present study, we only consider the demand for FP among married or in-union women aged 15–49 years.

FP2020% and 75% demand satisfied are two ambitious family planning initiatives. A recent assessment of FP2020 found that progress has been made with diverse country-level growth rates, but overall the initiative is below the proposed trajectory.[13] Given the scale of the initiatives and the number of partners involved in the family planning field, improved coordination and a broader coalition are necessary to achieve the goals. The objective of this study is to assess the concordance of these two initiatives by estimating the implication of accomplishing one target on the other. A demonstration of their consistency, or the lack thereof, provides a better understanding of the proposed quantitative goals and helps to formulate collective strategies.

## METHODS

The contraceptive prevalence data are from the United Nations Development Programme (UNDP) survey-based estimates of the percentage of married or in-union women aged 15–49 years using any modern contraceptive method.[14] The database includes estimates of modern contraceptive prevalence rate (mCPR) and % demand satisfied collected from 466 surveys in 142 countries from 1986 to 2016. Among the 70 FP2020 focus countries (South Africa joined the FP2020 Initiative after the London Summit), three countries (Djibouti, Somalia and Western Sahara) do not have any survey-based estimates of mCPR and % demand satisfied and therefore are excluded from the present study. In the end, our study is based on 67 FP2020 countries, with a focus on the 41 countries that made a commitment to the FP2020 Initiative (defined as commitment-making countries; see www.familyplanning2020.org for a full and up-to-date list; accessed on 20 February 2019).

The target measures discussed in this study are closely correlated by definition. Let P denote the total number of women aged 15–49 years, N denote the number of women who express a need for family planning, C denote the number of female modern contraceptive users, T denote the number of modern and traditional contraceptive users, U denote the number with unmet need for family planning. Then, we have mCPR=C/P, % unmet need=U/P and % met need (or demand satisfied)=C/N.

$$\% \ satisfied \ demand = \frac{C}{N} = \frac{C}{T+U} = \frac{mCPR}{CPR+\%unmet \ need}$$

An increase in C implies higher mCPR, but it does not necessarily increase % demand satisfied. The relationship between the indicators becomes complex in other scenarios, such as when more women express a need for family planning. This will decrease the % met need without affecting mCPR. The congruence, and lack of it, has been observed in FP2020 countries. From 2012 to 2017, the high growth of mCPR has driven a 9 percentage point increase in demand satisfied in Eastern and Southern Africa. During the same period, Central and West Africa experienced comparable mCPR growth, but that was accompanied by increasing levels of unmet need. These are the results of a complex dynamic involving both fertility intentions and available family planning services. As a result, our subsequent empirical analyses will be based on probabilistic statistical regression rather than deterministic mathematical relationships.

Another complicating factor is that FP2020 counts all women, irrespective of their marital status, while the 75% target only covers married or in-union women. Although subsequent debates consider expanding the demand satisfied target to all women, no consensus has been reached, and therefore, we will use the original statement of the 75% target. The difference in denominators will be dealt with in our statistical models.

The congruence between FP2020% and 75% demand satisfied targets requires a bi-directional assessment. We estimated the implications of achieving one of them on the other. Specifically, the study attempts to answer the following two questions: (1) how many additional users will be added following the 75% demand satisfied target; (2) what percentage of demand will be satisfied in 41 commitment-making countries assuming an annual increase of 1.4 percentage points from 2012 until 2030? Annual growth of 1.4% is the overall target proposed by the London Summit on Family Planning Metrics Group across all FP2020 focus countries.[9] Overall annual growth of 0.7 percentage points was observed across the world's 69 poorest countries before 2012. Brown et al[9] estimated that doubling the annual growth to 1.4 would add 120 million female modern contraceptive users by 2020. The target growth rate is considered an aspirational yet achievable goal assuming that the resources and leadership around current family planning programme may be collectively mobilised. These two assessments are conducted separately, although employing a similar methodology (figure 1).

There are three steps to answer the first research question. The first step is to estimate the necessary married-woman mCPR to satisfy 75% demand with modern methods by 2030. Among the 41 commitment-making countries, five FP2020 commitment-making countries had already reached the 75% demand satisfied goal in their most recent surveys (table 1). It is reasonable to assume that maintaining at least 75% demand satisfied by 2030 is the goal in those countries. We assume that the mCPR and % demand satisfied will remain at their most recent observed level until 2030.

**Table 1** Modern contraceptive prevalence rate (mCPR) in five commitment-making countries where % demand satisfied exceeded 75% in the last survey

| Country | Region | Survey date | mCPR | % demand satisfied |
|---|---|---|---|---|
| Myanmar | Non-SSA | 2015–2016 | 51.3 | 75.0 |
| Kenya | SSA | 2015 | 62.6 | 76.2 |
| Indonesia | Non-SSA | 2015–2016 | 59.5 | 78.8 |
| South Africa | SSA | 2003–2004 | 59.8 | 81.1 |
| Zimbabwe | SSA | 2015 | 65.8 | 85.2 |

Non-SSA includes all other regions.
SSA, sub-Sahara Africa.

For the other 36 countries, the percentage of demand satisfied with modern methods is assumed to reach 75% in 2030. Then, we employ the following country-level fixed effects longitudinal model to estimate the required mCPR for the assumed 75% demand satisfied.

$$y_{it} = \beta_0 + \beta_1 x_{it} + \beta_2 x_{it}^2 + \alpha_i + \varepsilon_{it} \quad (1)$$

where $y_{it}$ denotes the mCPR for country $i$ in time $t$; $x_{it}$ denotes the % demand satisfied for country $i$ in time $t$; $\alpha_i$ denotes the time-invariant unobserved fixed effects for country $i$; $\varepsilon_{it}$ denotes the error term. The mode is chosen from several options due to its best predictive performance. The model is first fitted using survey-based data compiled by the UN. The least squares dummy variable method is used in the model estimation.[15] This approach explicitly provides the coefficients of the country dummy, which is required in predicting the mCPR for the assumed 75% demand satisfied in 2030. Then with the estimated coefficients and country-level fixed effects, we estimate mCPR for the assumed 75% demand satisfied.

The second step is to convert the married-woman mCPR estimated in step 1 to all-woman mCPR. Two

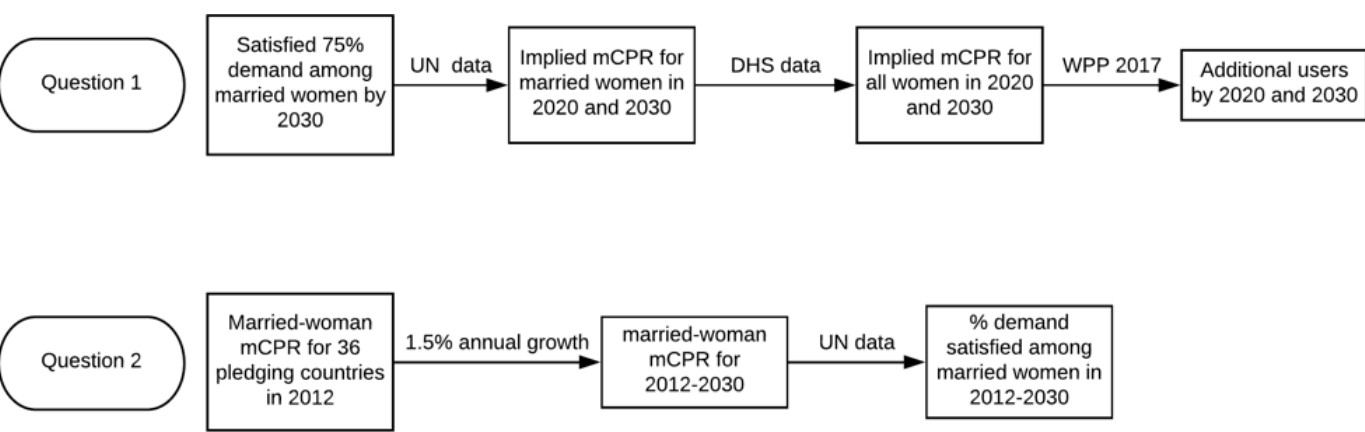

**Figure 1** Analytical flowchart for the two research questions. mCPR, modern contraceptive prevalence rate. (DHS: Demographic and Health Surveys; WPP: World Population Prospects)

hundred sixty-two Demographic and Health Surveys (DHS) based on samples of all women of reproductive ages were conducted from 1990 to 2016 in 85 countries. We use the following fixed effects longitudinal model to estimate all-woman mCPR from married mCPR:

$$a_i = \theta_0 + \theta_1 m_i + v_i + \epsilon_{it} \qquad (2)$$

where $a_i$ and $m_i$ denote the all-woman and married mCPR in survey $i$; $v_i$ denotes region (SSA vs non-SSA) level fixed effects. We use region level instead of country-level fixed effects because a model with country-level fixed effects cannot be used for prediction in FP2020 countries without a DHS survey.

In the third step, we assume that all-woman mCPR will increase linearly from the level in the last survey to the level estimated for 2030 in step 2. Using the number of women of reproductive age obtained from World Population Prospects 2017, we calculate the number of female modern contraceptive users in the 67 FP2020 focus countries.[16]

The second research question is answered similarly in three steps (figure 1). We first estimate the baseline, that is, all-woman mCPR in 2012. Our principle is to rely on the survey-based estimates as much as possible. As mentioned above, 5 of the 41 FP2020 commitment-making countries had already reached the 75% demand satisfied demand goal in their most recent surveys and therefore are excluded from this investigation. Among the other 36 commitment-making countries, 10 conducted a survey in 2012. For those 19 countries that have conducted surveys both before and after 2012, we use the two surveys before and after 2012 to linearly interpolate the mCPR for 2012. For the other seven countries that only have surveys conducted before 2012, we used the last survey-based estimate for 2012.

Then we impose a 1.4% annual increase in all-woman mCPR from 2012 until 2030. Finally, we predict the % demand satisfied demand associated with the calculated levels of all-woman mCPR for 2012–2030 based on a fixed effects longitudinal model similar to Equation (1), but moving % demand satisfied to the left-hand side and including mCPR and its squared term in the right-hand side.

### Patient and public involvement
The study does not involve patients or the public.

### RESULTS
All three fixed effects longitudinal models fit the data quite well, indicating excellent predictive performance (table 2). Using 466 survey-based estimates, the adjusted R-squared of the model regressing married-woman mCPR on % demand satisfied and a country dummy is above 0.98, meaning that less than 2% of the variations in married-woman mCPR cannot be explained by the model (Model 1). As a result, the estimated married-woman mCPR based on the assumed 75% demand satisfied should be highly accurate and reliable. The adjusted R-squared

**Table 2** Goodness of fit of the fixed effects longitudinal models

|  | Model 1 | Model 2 | Model 3 |
|---|---|---|---|
| Outcome | Married mCPR | All-woman mCPR | % satisfied demand |
| Covariates | % demand satisfied (% demand satisfied)^2 | Married mCPR | Married mCPR (married mCPR)^2 |
| Fixed effects | Country level | Region (SSA; non-SSA) level | Country level |
| Sample size | 466 | 262 | 466 |
| R-squared | 0.98 | 0.91 | 0.97 |

mCPR, modern contraceptive prevalence rate; SSA, sub-Sahara Africa.

of 0.97 in Model 2 also indicates accurate conversion from married-woman to all-woman mCPR. Model 3 that regresses % demand satisfied on married-woman mCPR also performed well (adjusted R-squared 0.97).

Achieving the 75% demand satisfied by 2030 goal means a gain of approximately 82 million additional users in these 67 FP2020 countries from 2012 to 2020, which is about 68% of the 120 million proposed by the FP2020 Initiative (table 3). From 2012 to 2020, these 41 commitment-making countries will contribute 74 million additional users while these 26 non-commitment-making FP2020 countries contribute 8 million. If the 67 countries continue the mCPR growth rate implied by the 75% demand satisfied initiative, the goal of adding 120 million female modern contraceptive users will be achieved in early 2023 (figure 2). By 2030, there will be 184 and 21 million additional users in commitment-making and non-commitment-making countries, respectively, making a total number of 206 additional modern contraceptive users in these 67 FP2020 countries.

Five of the 41 FP2020 commitment-making countries (three in sub-Saharan Africa) have already satisfied 75% or more of the contraceptive demand, according to their last survey. Among the other 36 commitment-making countries, only four additional countries (Bangladesh, India, Malawi and Vietnam) will reach that target by 2020, following FP2020's proposed 1.4% annual increase in mCPR (table 4). Another eight countries (Ethiopia, Laos, Madagascar, Nepal, Rwanda, Solomon Islands, Tanzania and Zambia) will do so by 2030. Disaggregated by region, the situation is more challenging in sub-Saharan Africa, where only one (Malawi) of the 26 commitment-making countries will reach the 75% target by 2020, and another five countries (Ethiopia, Madagascar, Rwanda, Tanzania and Zambia) will do so by 2030. Adding those three countries that had already reached the target in their most recent surveys, less than one-third (9) of the 29 commitment-making countries in this region will satisfy 75% demand for family planning by 2030. In the other

**Table 3** Modern contraceptive prevalence rate (mCPR), modern contraceptive users (thousand) and added users since 2012 (thousand) in 41 commitment-making and 26 non-commitment-making countries under FP2020: assuming the achievement of 75% demand satisfied by 2030

| Country | 2012 | | 2020 | | | 2030 | | |
|---|---|---|---|---|---|---|---|---|
| | mCPR | Users | mCPR | Users | Added users | mCPR | Users | Added users |
| Commitment-making countries (41) | | | | | | | | |
| Afghanistan | 10.4 | 691 | 22.7 | 2065 | 1374 | 38.1 | 4645 | 3954 |
| Bangladesh | 41.2 | 17800 | 42.4 | 20200 | 2366 | 43.8 | 22200 | 4369 |
| Benin | 7.0 | 158 | 22.0 | 636 | 478 | 40.9 | 1570 | 1412 |
| Burkina Faso | 13.5 | 515 | 25.5 | 1249 | 734 | 40.5 | 2704 | 2189 |
| Burundi | 16.7 | 371 | 27.5 | 768 | 397 | 41.0 | 1599 | 1229 |
| Cameroon | 14.2 | 708 | 25.8 | 1621 | 913 | 40.3 | 3362 | 2655 |
| Chad | 0 | 0 | 17.2 | 651 | 651 | 39.3 | 2052 | 2052 |
| Côte d'Ivoire | 11.5 | 566 | 24.2 | 1509 | 943 | 39.9 | 3277 | 2711 |
| DR Congo | 3.9 | 594 | 20.3 | 4066 | 3472 | 40.8 | 11600 | 11000 |
| Ethiopia | 26.2 | 5677 | 32.6 | 9285 | 3608 | 40.5 | 15100 | 9398 |
| Ghana | 14.3 | 935 | 26.9 | 2093 | 1158 | 42.6 | 4080 | 3145 |
| Guinea | 4.8 | 124 | 20.1 | 655 | 531 | 39.2 | 1711 | 1588 |
| Haiti | 24.5 | 663 | 32.9 | 1009 | 346 | 43.3 | 1493 | 829 |
| India | 35.7 | 115000 | 38.5 | 138000 | 22600 | 41.9 | 162000 | 46700 |
| Indonesia | 45.7 | 31000 | 45.7 | 32900 | 1904 | 45.7 | 34600 | 3521 |
| Kenya | 47.1 | 5076 | 47.7 | 6592 | 1516 | 48.5 | 8575 | 3499 |
| Laos | 33.4 | 570 | 38.1 | 742 | 171 | 44.1 | 972 | 401 |
| Liberia | 14.2 | 139 | 25.9 | 318 | 179 | 40.6 | 649 | 510 |
| Madagascar | 26.2 | 1395 | 32.8 | 2246 | 851 | 41.0 | 3674 | 2280 |
| Malawi | 45.4 | 1696 | 44.7 | 2218 | 522 | 44.0 | 2984 | 1289 |
| Mali | 7.6 | 268 | 21.2 | 962 | 694 | 38.1 | 2427 | 2160 |
| Mauritania | 7.6 | 70 | 22.2 | 259 | 189 | 40.5 | 610 | 540 |
| Mozambique | 11.3 | 676 | 23.3 | 1792 | 1115 | 38.2 | 4002 | 3325 |
| Myanmar | 39.5 | 5503 | 39.5 | 5956 | 453 | 39.5 | 6197 | 694 |
| Nepal | 34.0 | 2504 | 38.1 | 3293 | 789 | 43.4 | 4078 | 1574 |
| Niger | 5.2 | 192 | 19.2 | 971 | 778 | 36.6 | 2781 | 2588 |
| Nigeria | 6.2 | 2364 | 20.5 | 9773 | 7409 | 38.5 | 24200 | 21900 |
| Pakistan | 20.0 | 8935 | 29.5 | 15400 | 6475 | 41.3 | 26100 | 17100 |
| Philippines | 28.1 | 7022 | 35.8 | 10100 | 3100 | 45.5 | 14700 | 7664 |
| Rwanda | 36.1 | 974 | 39.2 | 1326 | 353 | 43.1 | 1880 | 906 |
| Senegal | 12.8 | 429 | 24.7 | 1040 | 611 | 39.6 | 2217 | 1787 |
| Sierra Leone | 11.6 | 188 | 23.7 | 479 | 291 | 38.8 | 997 | 809 |
| Solomon Islands | 24.1 | 33 | 28.0 | 45 | 13 | 32.7 | 64 | 31 |
| South Africa | 46.4 | 6715 | 46.4 | 7366 | 651 | 46.4 | 8067 | 1352 |
| South Sudan | 6.3 | 160 | 21.2 | 697 | 537 | 39.9 | 1715 | 1555 |
| Togo | 11.8 | 195 | 25.1 | 516 | 322 | 41.7 | 1109 | 915 |
| Uganda | 20.5 | 1652 | 30.7 | 3355 | 1703 | 43.4 | 6715 | 5062 |
| Tanzania | 21.8 | 2483 | 30.1 | 4459 | 1976 | 40.4 | 8243 | 5760 |
| Vietnam | 43.4 | 11200 | 45.8 | 11900 | 686 | 48.8 | 12600 | 1401 |
| Zambia | 34.3 | 1176 | 38.0 | 1731 | 555 | 42.7 | 2627 | 1451 |
| Zimbabwe | 50.9 | 1944 | 50.9 | 2366 | 422 | 50.9 | 3010 | 1066 |

Continued

**Table 3** Continued

| Country | 2012 | | 2020 | | | 2030 | | |
|---|---|---|---|---|---|---|---|---|
| | mCPR | Users | mCPR | Users | Added users | mCPR | Users | Added users |
| Subtotal | | 238350 | | 312597 | 73836 | | 423176 | 184372 |
| Non-commitment-making countries (26) | | | | | | | | |
| Bhutan | 50.2 | 101 | 50.2 | 114 | 14 | 50.2 | 124 | 23 |
| Bolivia | 30.5 | 788 | 37.6 | 1129 | 341 | 46.6 | 1590 | 801 |
| Cambodia | 28.5 | 1159 | 35.1 | 1581 | 422 | 43.3 | 2281 | 1121 |
| Central African Republic | 12.7 | 133 | 24.7 | 292 | 158 | 39.7 | 627 | 493 |
| Comoros | 12.0 | 21 | 25.3 | 54 | 33 | 41.9 | 114 | 92 |
| Congo | 10.9 | 120 | 24.9 | 334 | 214 | 42.3 | 755 | 635 |
| Egypt | 43.8 | 9961 | 43.8 | 11200 | 1256 | 43.8 | 13200 | 3262 |
| Eritrea | 9.9 | 108 | 23.1 | 311 | 203 | 39.6 | 704 | 595 |
| Gambia | 5.6 | 24 | 19.8 | 109 | 85 | 37.6 | 282 | 258 |
| Guinea-Bissau | 10.2 | 41 | 18.1 | 90 | 49 | 27.9 | 180 | 139 |
| Honduras | 49.0 | 1104 | 49.0 | 1319 | 215 | 49.0 | 1495 | 391 |
| Iraq | 28.5 | 2270 | 33.5 | 3367 | 1097 | 39.6 | 5157 | 2887 |
| Kyrgyzstan | 26.3 | 401 | 32.1 | 501 | 100 | 39.4 | 703 | 301 |
| Lesotho | 46.4 | 251 | 46.4 | 290 | 39 | 46.4 | 337 | 86 |
| Mongolia | 36.9 | 305 | 39.5 | 331 | 26 | 42.8 | 392 | 87 |
| Nicaragua | 59.2 | 960 | 59.2 | 1053 | 93 | 59.2 | 1119 | 159 |
| North Korea | 58.5 | 3899 | 58.5 | 3800 | −100 | 58.5 | 3654 | −246 |
| Palestine | 33.1 | 348 | 37.0 | 490 | 143 | 41.9 | 717 | 369 |
| Papua New Guinea | 24.6 | 449 | 32.5 | 717 | 267 | 42.3 | 1130 | 680 |
| Sao Tome and Principe | 27.3 | 12 | 36.2 | 19 | 7 | 47.4 | 32 | 20 |
| Sri Lanka | 42.1 | 2251 | 44.6 | 2374 | 123 | 47.6 | 2473 | 222 |
| Sudan | 6.5 | 552 | 21.1 | 2240 | 1688 | 39.3 | 5425 | 4873 |
| Tajikistan | 20.3 | 428 | 28.0 | 670 | 242 | 37.5 | 1086 | 657 |
| Timor-Leste | 18.2 | 46 | 24.0 | 74 | 28 | 31.1 | 127 | 81 |
| Uzbekistan | 39.9 | 3334 | 40.1 | 3597 | 263 | 40.3 | 3964 | 630 |
| Yemen | 21.8 | 1326 | 30.3 | 2339 | 1013 | 41.0 | 4079 | 2753 |
| Subtotal | | 30395 | | 38397 | 8019 | | 51744 | 21372 |
| Total (67 focus countries) | | 268745 | | 350995 | 81855 | | 474920 | 205744 |

The predicted mCPR for Chad 2012 was rounded to 0; the columns may add up exactly because our statistical models used exact numbers while results are presented in thousands.

regions, five countries will achieve the target by 2020, and another three will do so by 2030. Those eight target-achieving countries represent two-thirds of the 12 non-SSA commitment-making countries.

In sum, assuming FP2020's proposed annual growth rate in mCPR, the % satisfied will reach 75% in less than half (17) of the 41 FP2020 commitment-making countries.

## DISCUSSION

The contribution of this study is an improved understanding of the concordance of two global family planning initiatives: FP2020's adding 120 female modern contraceptive users by 2020 in 69 of the world's poorest countries and USAID's satisfying 75% demand for family planning with modern contraceptives. Our results show that the two initiatives move towards the same goal of promoting access to family planning for women and girls. Overall, both the 75% demand satisfied and the FP2020 goal are ambitious. Achieving the 75% demand satisfied goal by 2030 implies that 82 million or 68% of the 120 million target users will be added by 2020 in 67 FP2020 focus countries. The target of 120 million will be achieved by

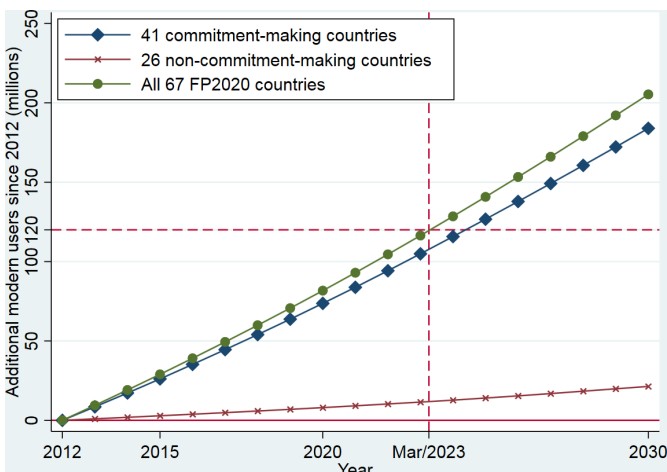

**Figure 2** Number of additional female modern contraceptive users in 41 commitment-making and 26 non-commitment-making countries assuming the trajectory of satisfying 75% demand by 2030.

2023, only 3 years later than the FP2020 deadline. On the contrary, achieving a 1.4% annual increase in all-woman mCPR will enable only 17 of the 41 commitment-making countries to attain the goal of 75% demand satisfied by 2030. The overall assessment should not mask the across-country variations. In some countries, it is more plausible to achieve the FP2020's proposed annual increase of 1.4 percentage points than satisfying the 75% demand by 2030. Some other countries, however, have 75% demand satisfied or will do so by 2030 with an annual mCPR increase below 1.4 percentage points.

Capitalising the shared goals, the demonstrated concordance may facilitate building a broad coalition to promote family planning in the developing world. These two initiatives represent the objectives of two major donors to family planning.[17] The % demand satisfied has also been adopted as an indicator of the sustainable development goals. Due to their different features and advantages, mCPR (and number of users) and % demand satisfied will continue coexisting in the international agenda for family planning. Despite their theoretical correlation, the empirical relation between the two indicators depends on other context-specific factors, such as demand generation and changes in fertility desire. As a result, an assessment of the empirical correlation between the two indicators has sustaining policy implications. A consensus goal is critical to building a broad coalition to collectively and effectively mobilise financial and political resources and capture global attention.

The simulated implications of achieving one target on the other have several policy implications, which are urgently needed as donors and stakeholders are debating about the post-FP2020 plan. First, multiple measures will continue coexisting in international family planning. The FP2020 Core Group of which the BMGF, United Kingdom's DFID, USAID and United Nations Population Fund (UNFPA) are active may renew their commitment to adding female modern contraceptive users beyond

**Table 4** Modern contraceptive prevalence rate (mCPR) and % demand satisfied (%SD) in 2012, 2020 and 2030 in 36 commitment-making countries: assuming the achievement of FP2020 and extending its mCPR trajectories to 2030

| Country | 2020 | | 2030 | |
|---|---|---|---|---|
| | mCPR | %SD | mCPR | %SD |
| Sub-Sahara Africa (26) | | | | |
| Benin | 19.1 | 33.7 | 33.1 | 52.0 |
| Burkina Faso | 27.4 | 47.1 | 41.4 | 63.5 |
| Burundi | 28.9 | 46.5 | 42.9 | 62.5 |
| Cameroon | 27.8 | 48.3 | 41.8 | 64.5 |
| Chad | 15.9 | 31.4 | 29.9 | 50.5 |
| Côte d'Ivoire | 23.7 | 42.6 | 37.7 | 59.8 |
| DR Congo | 18.8 | 33.5 | 32.8 | 51.9 |
| Ethiopia | 40.9 | 63.6 | 54.9 | **76.7** |
| Ghana | 32.8 | 50.5 | 46.8 | 65.6 |
| Guinea | 15.8 | 31.8 | 29.8 | 50.9 |
| Liberia | 28.9 | 49.0 | 42.9 | 65.0 |
| Madagascar | 40.4 | 61.4 | 54.4 | **74.7** |
| Malawi | 62.1 | **77.7** | 76.1 | **85.8** |
| Mali | 21.1 | 42.6 | 35.1 | 60.5 |
| Mauritania | 22.7 | 40.8 | 36.7 | 58.3 |
| Mozambique | 22.5 | 45.6 | 36.5 | 63.1 |
| Niger | 23.4 | 49.9 | 37.4 | 67.2 |
| Nigeria | 22.0 | 44.7 | 36.0 | 62.3 |
| Rwanda | 56.8 | 73.3 | 70.8 | **82.6** |
| Senegal | 27.3 | 48.6 | 41.3 | 65.0 |
| Sierra Leone | 24.9 | 47.9 | 38.9 | 64.9 |
| South Sudan | 12.9 | 23.8 | 26.9 | 43.6 |
| Togo | 26.9 | 42.8 | 40.9 | 59.3 |
| Uganda | 37.0 | 54.6 | 51.0 | 68.7 |
| Tanzania | 40.5 | 63.8 | 54.5 | **77.0** |
| Zambia | 53.1 | 72.0 | 67.1 | **82.2** |
| Other regions (10) | | | | |
| Afghanistan | 31.0 | 56.9 | 45.0 | 72.4 |
| Bangladesh | 70.5 | **83.1** | 84.5 | **89.2** |
| Haiti | 42.5 | 59.7 | 56.5 | 72.5 |
| India | 59.1 | **78.0** | 73.1 | **86.8** |
| Laos | 53.9 | 70.1 | 67.9 | **80.2** |
| Nepal | 54.4 | 72.1 | 68.4 | **82.0** |
| Pakistan | 37.3 | 57.7 | 51.3 | 71.6 |
| Philippines | 48.3 | 62.7 | 62.3 | 74.0 |
| Solomon Islands | 38.6 | 73.3 | 52.6 | **87.0** |
| Vietnam | 69.7 | **77.3** | 83.7 | **83.6** |

Bold indicates reaching the target of satisfying 75% demand for family planning; Madagascar's 74.7% in 2030 can be rounded to 75%.

FP2020 to the FP2030 deadline. The % demand satisfied has been adopted as an indicator, Sustainable Development Goals (SDG) 3.7.1. The 75% benchmark is being used as a proxy for the minimum definition of 'universal access to reproductive health' in terms of contraceptive

use (SDG 3). Methodologically, our models for assessing the congruence of the two measures could be replicated as the FP2020 movement sets its goals for FP2030.

Second, our exercise sheds light on the choice between aspirational and realistic target-setting approaches. The findings show that 75% demand satisfied can be viewed in three settings: (1) countries who have already achieved the goal but whose plans involve increasing the percentage higher than the 75% benchmark (eg, Indonesia, Myanmar, Kenya, South Africa, Zimbabwe); (2) countries which are projected to likely reach the goal by 2020 and 2030 and (3) countries which will remain below the goal (24 of 41 commitment-making countries). With only 1 year left before its deadline, FP2020 has contributed to the mobilisation of global resources for family planning and has shown progress against the goal but not at the trajectory to reach 120 million more women and girls by 2020.

The third policy implication is for the choice between global and national targets. All countries in our exercise belong to low-income countries, but they still demonstrate massive diversity in terms of mCPR, desired and realised fertility, and population age structure. When setting targets in the future, donors and stakeholders need to strike a balance between simplification (global target as in FP2020) and customisation (country-specific targets as in 75% demand satisfied).

The last policy implication is on SDG. Although % demand satisfied has been adopted as an indicator (SDG 3.7.1), it has not been associated with quantitative goals. The same situation occurred to Target 5b of Millennium Development Goals (MDGs): 'Achieve, by 2015, universal access to reproductive health'. Several studies argued that clear, measurable goals could be a focal point for coalescing political support for action.[9] Reflecting on the lag in substantively integrating family planning into the MDGs, FP2020 proposed a quantifiable target of adding 120 million female modern contraceptive users by 2020. Adopting the target of 75% demand satisfied in SDG may help mobilise and guide resource allocation and provide a benchmark for programme advocacy. Per our simulation results, the target is achievable in certain countries and aspirational in others.

The study is not without limitations. First, despite the highly satisfactory model fit, our regressions could be theoretically improved by including other factors such as calendar time. We did not include year as a covariate because its coefficient reflects not only temporal effects but also the changing composition of countries in the database. For example, the earliest DHS surveys were mostly in Africa, while Asia was added later. So, the absence of calendar time in the model is a limitation with the database rather than our methodology. Since we are mainly interested in the predictive performance of the model, measured by the adjusted R-squared, and adding year as a covariate changed the adjusted R-squared by less than 1 percentage point, our final model did not consider calendar time. The second limitation is the

linear assumption on mCPR growth. Other growth curves (such as S-shaped or logistic) may be more accurate in many countries. The 67 FP2020 countries are in different stages of mCPR growth, some experiencing a convex trajectory and some a concave trajectory. Fully accounting for country-specific curves will likely make the statistical models much more complex and less robust. We believe a linear trajectory provides an acceptable approximation for the mixture of convex and concave trajectories. Consequently, the global estimates presented in the study may not be substantially affected by the assumed linearity.

As repeatedly emphasised in the London Summit document, setting a quantitative target should not cause concern among those firmly committed to sexual and reproductive health and rights because all interventions will have women's rights at the centre of their implementation efforts. Our assessment in this study of the congruence of major, articulated family planning initiatives aims to unite international communities into collective actions that secure women's and girls' access to effective contraceptive methods.

**Contributors** QL, SA and JGR devised the study and wrote the article. QL compiled the data and led the statistical modeling and analysis.

**Funding** This study was funded by Bill and Melinda Gates Foundation.

**Competing interests** None declared.

**Patient consent for publication** Not required.

**Provenance and peer review** Not commissioned; externally peer reviewed.

**ORCID iD**
Qingfeng Li http://orcid.org/0000-0002-6390-6921

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
