## [Reviewer comments · BMJ Open]

ARTICLE DETAILS

TITLE (PROVISIONAL)	Capitalizing on shared goals for family planning: a concordance assessment of two global initiatives using longitudinal statistical models
AUTHORS	Li, Qingfeng; Rimon, Jose; Ahmed, Saifuddin

VERSION 1 – REVIEW

REVIEWER	Kirsten Black The University of Sydney
REVIEW RETURNED	14-May-2019

GENERAL COMMENTS	Thank you for the opportunity to review this unique paper. This has great potential as an interesting manuscript but I suggest the following issues should be addressed to enhance it. Abstract Objectives: name the targets that are to be compared Methods: state the organisation leading the 75% target Results: The presentation of the results needs greater clarity. Start by clearly stating the difference in achieved goals of the two strategies and name them (rather than saying 'the latter') Conclusions; The conclusion needs to align with the results - currently it is just a motherhood statement. How are the two strategies complementary/overlapping Introduction page 4 second last sentence: the word 'more' needs to be inserted before controversial Methods: I found the methods section confusing in parts, particularly the section which differentiates between married all-women mCPR and all women all-women mCPR (page 8 line 35-40). Statistical review would be valuable. Results The First paragraph of the results is well tabulated and could be more briefly summarised so as to focus on the two strategies. Syntax errors with changes in verb tenses throughout the results need to be addressed. typo 'married' (page 10 line 45); the word million missing (page 11 line 2); incomplete sentence page 11 line 45 'another 9 countries...' Discussion The discussion is not strong and deserves more thought and depth. How are these two strategies being realised? Are they competing in their approach. Which is realistic? How should a broad coalition work as suggested in the abstract?
---

REVIEWER	Anne Pfitzer Jhpiego, United States
-----------------	--

	My NGO, Jhpiego, is affiliated with Johns Hopkins University I am currently collaborating with the last author on a completely separate study. However, I don't believe this interferes with my ability to provide an objective review.
REVIEW RETURNED	24-May-2019

GENERAL COMMENTS	General comment I read your paper with great interest and am convinced it will receive attention when published. On the eve of 2020 and as the FP global community reflects what should be our priorities for the coming decade, it is appropriate to reflect critically on past efforts and implications for the future. This study is indeed very timely. As noted at the end of your discussion, despite a leadership role of DFID, and joint efforts to engage USAID and UNFPA in preparing the 2012 London Summit, the FP2020 agenda is closely associated with the Bill & Melinda Gates Foundation and with Melinda herself, a reflection of her determination to empower women and girls to have the number of children they want and when they want them. With the change in administration of the US Government, USAID has been perceived as distancing itself from the FP2020 agenda. This is perhaps signaled by the shaping of a strategy for 2030, and a new indicator and target. With this article, it appears that the authors are reflecting on the dynamics of what are effectively the two powerhouses of the global FP movement: USAID which still is a dominant donor for this work, and the Gates Foundation which has sought to catalyze action and be a strong champion for transforming the approaches of the FP community. By highlighting the divergence in indicators and targets, the authors seem to be making a political comment couched in sophisticated statistics. However, their conclusion is a bit lost. The argument seems to be that the 75% demand satisfied goals will achieve results faster than numerical targets underpinned by setting goals related to annual percentage increases (I think the original FP2020 metrics group adjusted the annual increase based on recent country performance, which has been somewhat glossed over in this study, some of which is defensible for the clarity of the analysis). However, is that the best course of action? There are important limitations to a complex indicator such as the demand met with modern contraception one (not unlike the unmet need indicator). I hope the authors can sharpen their own position and I look forward to the ensuing debates. My chief criticism of the entire approach to this study is that it fails to acknowledge the recent recognition and consensus that mCPR growth is not typically linear in nature. In fact, recent FP2020 metrics experts have determined that country family planning growth patterns follow an S-curve (see FP2020 latest progress report at http://progress.familyplanning2020.org/content/measurement). Admittedly, I was not able to find any publication in the scientific, peer-reviewed literature to support these assertions. Yet, all the countries that have experienced bursts of mCPR growth in recent years (first Malawi, Rwanda and Ethiopia and more recently Mozambique, Liberia, Malawi again and Kenya), have been in this middle range of prevalence, in support for the S-curve agreement. The architects of the FP2020 goal (the authors cited the Brown et al 2014 paper) constructed the 120 million additional users goal using country projections assumed a linear trend in mCPR growth at the
---

	time. I have heard some acknowledge that a major flaw in the calculation of the FP2020 target was the failure to recognize and accommodate for the S-curve pattern. As a result, the target assigned for growth in India, the country with the largest population in the FP2020 list, was definitely overly ambitious and had a disproportional effect on the overall target. If we could go back in time to reduce the target and align it with a more realistic, yet ambition target for India, then the numerical goal would perhaps not have been too ambitious, given that the bulk of FP2020's 69 countries are still in the early or middle stages of the S-curve. All this to say that the authors could potentially reference the lessons imparted by the S-curve in their paper, which to my mind, has dramatically transformed target setting for countries developing FP strategies. The next implication of this is that if, as Cahill et al found, the global mCPR is 45.7%, then the world is approaching the slower part of the S-curve, which may have a dragging effect on achieving either target by 2030. But perhaps this is for another paper. More detail-oriented, specific comments: Abstract (and introduction) The authors characterize family planning as a medical intervention. I suggest you replace term "medical" with "health" as medical implies that a physician has to provide it and some methods, such as fertility awareness methods, have no medical component to them at all, yet arguably impart health benefits. Nowhere in the abstract is the 2nd global initiative named. When I read it, I recognized it or assumed I recognized it as the Sustainable Development Goals (whose indicator USAID has adopted for its own strategy). I suggest you include greater clarity about who "owns" this target in the text and clarify which initiative you mean in the abstract. As it was only when I got to page 7 or 8 that I realized you were talking about USAID's strategy. Indeed, the target is USAID's as the SDG target is phrased more broadly even if the indicator is common across the two initiatives. Anyway, this vagueness is an issue in my view. Results section of abstract uses the word "latter": Latter is not clear here. The goals are listed in order of FP2020 and SDGs in the methods section. Why not be explicit and say FP2020? Introduction The summation of the Cahill et al conclusion that the FP2020 goals were overambitious somewhat misrepresents the nuance of what that paper seemed to convey. I believe they allude to the diversity of countries and the demographic pull of large countries on the targets. Data You explain that the analysis includes data from 67 countries (page 6, line 49, 1st instance – appears again in Methods section (age 9, line 27), yet in the Results (page 11, line 8, ist instance), you refer to 66 countries. What happened to the missing country? Results Incomplete sentence on page 11, lines 42-45. Limitations
--	--

	The strong fit of the model may imply that I am wrong to make this comment, however, I wonder if the authors can comment on the linearity of the modeling assumptions as opposed to the typical patterns in mCPR growth. Discussion Page 13, second paragraph. The authors write: “On the other hand, achieving a 1.5% annual increase in all-woman mCPR will enable less than half of the 41 pledging countries to attain the goal of 75% satisfied demand by 2030.” I am troubled by the over-simplification of the target-setting that this sentence implies. I don’t think that this how Brown et al described the target setting for the FP2020 target or how countries go about setting targets. Authors should acknowledge that the 1.5% measure is one that they adopted as a convenience for their study. And that it doesn’t pass the test for either global or country targets without understanding the stage a country is in and recent patterns of growth, the demographic or youth bulge that it must also deal with. Conclusion statement in abstract. I missed this statement in the discussion or a strong link back to the title of the paper, so we are left with the somewhat weak “conclusion” in the abstract. Are the authors arguing for further convergence in the post-2020 period? The statement implies that the community should mobilize around both metrics. Given the title of the paper uses the term “shared goals”, I was hoping the authors would take more of a position given the critical timing of this paper. Tables and figures Table 1 and 3: I suggest replacing the term “pledging” with the more common language of “commitment-making” countries Table 3: The title of this table doesn't explain well what data are being presented here. If I understood correctly, the 2020 and 2030 data are estimates assuming either achievement of 75% demand satisfied by 2030. I inferred this from the text. However, data in tables should stand alone and be interpretable without reading the text, thus the title should be made clearer as to the analysis performed to arrive at these estimates. Table 4. Similar comment as Table 3 Figure 2: I suggest you add a horizontal line at 0, or move the vertical legend so that 0 is at the corner, otherwise it visually looks like the non-commitment-making countries are contributing more than the space (difference) between the commitment-makers and the total.
--	---

REVIEWER	Asad Khan The University of Queensland Australia
REVIEW RETURNED	03-Jul-2019

GENERAL COMMENTS	Insufficient information about statistical methods and data.
--

REVIEWER	Tesfalide Tekelab University of Newcastle, Australia and Wollega University, Ethiopia
REVIEW RETURNED	07-Jul-2019

GENERAL COMMENTS	I found the manuscript interesting. It is important topic. I have some
--

	minor comments. 1. The conclusion is not written clearly. I have seen only the discussion part. Write the conclusion inline with your finding 2. Line 42-45 – Needs revision. The sentence is not complete. “Another 9 countries (Ethiopia, Laos, Madagascar, Nepal, Philippines, Rwanda, Solomon Islands, Tanzania, and Zambia)” 3. Line 52- Change the word “discus” to discuss.
--	--

REVIEWER	Andrew Hinde University of Southampton, United Kingdom
REVIEW RETURNED	09-Jul-2019

GENERAL COMMENTS	Building a coalition to promote family planning through shared goals: assessing the concordance of two initiatives This paper examines the concordance between two family planning initiatives: (1) the addition of 120 million contraceptive users by 2020, and (2) satisfying 75 per cent of the demand for modern contraception among married or in-union women aged 15-49 years by 2030. It seems an interesting exercise to assess the implied consistency between the objectives of these initiatives, for there should be some relationship between them. Indeed, one of my suggestions is that you explore the theoretical relationship between them before embarking on your empirical analysis. Unfortunately, the paper as it stands is hard to follow and has some serious weaknesses. In my opinion it requires substantial revision before it could be published. I have five general points to make and a number of specific issues with individual passages or sections. General points 1. My first point is very basic. Do the 120 million contraceptive users to be added include males as well as females? If female, are they supposed to be married or in-union, or do any women who are persuaded to start using contraception count towards the target? (Actually, I know that the 120 million only includes women, but you might state this, for if they do include males much of the basis of your paper is undermined.) 2. Before embarking on your empirical analysis, it might be worth a brief theoretical exploration of the relationship between the various quantities you describe in the paper, taking the case of a single country. This will help you define quantities clearly, and establish in the minds of readers the difference between them. For example, if the number of (married or in-union?) women aged 15-49 years using modern contraception is C, the number of women who are not using but who do not wish to become pregnant at the present time is U, and the number who are trying to become pregnant or who do not wish to use contraception for other reasons is N, then the proportion of
---

demand satisfied is $C/(C + U)$. The prevalence rate is $C/(C + U + N)$. Of course, you have data for many countries, and hence you use statistical methods to establish the 'average' relationship across these countries. But the results should still not be too different from the theoretical relationships.

3. The theoretical excursion mentioned above would help you sort out one of the main weaknesses of the paper, which is the ambiguous or unclear definition of quantities. On p. 5, ll. 17-21, for example, there is confusion between 'all women' and 'women in a sexual union'. The paper refers to the percentage of demand satisfied as the 'proportion of all women who use modern contraception divided by the total demand for family planning' (ll. 18-19) but then defines the total demand for family planning as equal to the sum of the 'percentage of married or in-union women aged 15-49 who are using any contraception' (ll. 19-20) and the 'percentage of all women with unmet need' (l. 20). I was lost at this point. Can you define quantities and express what you mean precisely? The quantities should be defined in numbers of women, and you should clearly state in each case the age range you are considering, and whether you are restricting attention to married or in-union women.

4. As I understand your method, you use past data to estimate some coefficients. You use the latest contraceptive prevalence rate as a baseline and predict the contraceptive prevalence in 2030 using your estimated coefficients. This involves a big assumption that your coefficients will not change between now and 2030. On p. 8, ll. 11-12 you suggest that you will be predicting out of the range of your data (you will be using your model to make out of sample predictions). I think you need to do more to persuade me (and the reader) that you have accounted for this requirement when estimating the model. How do you assess the predictive performance of the model? What methods did you use? Did you use cross-validation, for example leave one out validation? On p. 12,

ll. 11-12 you say that you are 'mainly interested in the predictive performance of the model measured by the adjusted R-squared'. Predictive performance would be better assessed using cross-validation and the mean square error or the Akaike Information Criterion than by just using the *R*-squared (even though it is adjusted) on the model fit to past data.

5. The paper's conclusion, that 'a broad coalition needs to be formed to accomplish both initiatives' (p. 3, ll. 3-4) is hardly earth-shattering. I could have written that without needing all your analysis. What is new in your paper that needs emphasising? My take on your results is that the 75 per cent of demand satisfied goal is considerably more ambitious than just adding 120 million new users. This is worth emphasising.

Specific points

	p. 4, l. 20 Insert 'more' after 'planning is'. p. 5, l. 11 'priorities are' should be 'priorities is'. p. 5, l. 21 'any contraception' should, I think, be 'any modern contraception'. p. 5, l. 21 How do you measure 'unmet need'? This is an important issue for your paper, so could you explain how it is defined and calculated? p. 6, ll. 13-4 What are these 466 surveys? Are they Demographic and Health Surveys (DHSs), Multiple Indicator Cluster Surveys (MICS), or other surveys? How many of them are DHSs or MICS? p. 7, l. 6 'assuming a 1.5% annual increase'. Do you mean 1.5% or 1.5 percentage points? See also p. 10, l. 1. p. 7, ll. 18-19 It seems unnecessarily conservative to assume that the contraceptive prevalence rate and the percentage of demand satisfied will remain constant until 2030 for these well provided countries. p. 8, ll. 2-3 'For the other 36 countries, the percentage of FP demand satisfied with modern methods will reach 75% in 2030'. How do you know? Is this an assumption, and how does this relate to the 1.5% (or 1.5 percentage point) increase mentioned on p. 7, l. 6. p. 8, l. 8 'The mode is chosen from serval options' should be 'The model is chosen from several options'. p. 8, ll. 15-18 I presume that the reason you have to use two steps in this stage is that you do not have DHS data for all your 36 countries. Or, more accurately, 204 out of your 466 surveys are not DHSs based on samples of all women. If you used only the 262 DHSs based on samples of all women, you could do all this in one step, for the DHS data would allow you to compute the contraceptive prevalence for married and in- union women as well as all women. Can you explain why it is better to use the extra 204 surveys even though it makes the whole process more complicated
--	--

	and possibly less accurate? pp. 8-9 The symbols in the equations do not always mean the same thing. In equation (1) (p. 8, l. 5), yit refers to the contraceptive prevalence rate (CPR), and xit is the percentage of demand satisfied. However in equation (2) xi is the married and in-union CPR and yi is the all-women contraceptive prevalence rate. This is confusing for the reader. A symbol should mean the same thing throughout your paper. On p. 10, ll. 3-4 you say you are 'reversing the meaning of yit and xit'. Do not do this! Keep the meaning of the symbols the same and change the equation. p. 9, ll. 12-14 Why did you exclude the five countries who had reached the 75 per cent demand satisfied goal? They contribute 100 per cent achievement of the goal. p. 10, l. 7 Change 'involvd' to 'involved'. p. 10, ll. 12-13 'less than 2% of the variations in all-woman mCPR cannot be explained by the model'. This is very high. It suggests that contraceptive prevalence and the percentage of demand satisfied are very closely related. Does this not immediately suggest that there is a high degree of concordance between the two? p. 11, ll. 5-7 'Following the trajectory of increasing mCPR and % satisfied demand, the goal of adding 120 million modern contraceptive users will be achieved in early 2023'. I do not understand this sentence. p. 11, ll. 15-16 'Another 9 countries ...' will do what? p. 18, Table 3. There are several issues with this table. (1) How can Chad have a contraceptive prevalence rate that is negative? (2) The figures in for Bangladesh seem to be to be inconsistent: $17,800 + 2,366 \neq 20,200$ and $17,800 + 4.369 \neq 22,200$. (3) Have you considered comparing your married/in-union contraceptive prevalence rates estimated for 2020 with those for the latest DHSs for those countries where recent DHSs have been held? This is not difficult or time-consuming to do. Actually it is so quick that I did it in 6 minutes for the first 10 countries on your list using the Statcompiler on the DHS web site. Generally, for those countries with a DHS since 2015 the latest modern contraceptive prevalence rate is consistent with your 2020 predictions, though there are exceptions. For example, Ethiopia's modern contraceptive prevalence was 35 per cent in 2016, suggesting that your prediction of 32.6 per cent by 2020 is too low. You could do this kind of comparison for the predictions for 2020 in Table 3, p. 18 and Table 4, p. 21, to see how far adrift each country is likely to be. Of course, you cannot do it for all countries, but you can do it for several countries with DHSs in 2016, 2017 or 2018.
--	--

Responses to Reviewer 1's Comments

Reviewer Name: Kirsten Black

Institution and Country: The University of Sydney

Please state any competing interests or state 'None declared': None declared

Please leave your comments for the authors below

Thank you for the opportunity to review this unique paper. This has great potential as an interesting manuscript but I suggest the following issues should be addressed to enhance it.

Thank you for reviewing our manuscript and providing the valuable comments.

Abstract

Objectives: name the targets that are to be compared

We have revised that paragraph to make the comparison clear.

Methods: state the organisation leading the 75% target

United States Agency for International Development (USAID) proposed this target and is currently leading the initiative. Sustainable Development Goals later adopted this indicator as SDG 3.7.1. We have added this information to the manuscript.

Results: The presentation of the results needs greater clarity. Start by clearly stating the difference in achieved goals of the two strategies and name them (rather than saying 'the latter')

We have accordingly revised this paragraph to improve clarity.

Conclusions; The conclusion needs to align with the results - currently it is just a motherhood statement. How are the two strategies complementary/overlapping

We have added the implications of our statistical exercise on the concordance of the two strategies.

Introduction

page 4 second last sentence: the word 'more' needs to be inserted before controversial

Done. Thanks for the suggestion.

Methods: I found the methods section confusing in parts, particularly the section which differentiates between married all-women mCPR and all women all-women mCPR (page 8 line 35-40). Statistical review would be valuable.

Sorry for the confusion. We have added more details to the Methods section.

Results

The First paragraph of the results is well tabulated and could be more briefly summarised so as to focus on the two strategies.

While we consider it important to provide a detailed description of model performance, we agree that the paragraph overlaps with Table 2, and therefore have slightly abbreviate it.

Syntax errors with changes in verb tenses throughout the results need to be addressed.

Sorry for the errors. We have thoroughly proofread all sentences in the revision process.

typo 'married' (page 10 line 45); the word million missing (page 11 line 2); incomplete sentence page 11 line 45 'another 9 countries...'

We have corrected the errors.

Discussion

The discussion is not strong and deserves more thought and depth. How are these two strategies being realised? Are they competing in their approach. Which is realistic? How should a broad coalition work as suggested in the abstract?

We have substantially expanded the Discussion section to address the important issues you raised.

Responses to Reviewer 2's Comments

Reviewer Name: Anne Pfitzer

Institution and Country: Jhpiego, United States

Please state any competing interests or state 'None declared': My NGO, Jhpiego, is affiliated with Johns Hopkins University I am currently collaborating with the last author on a completely separate study. However, I don't believe this interferes with my ability to provide an objective review.

Please leave your comments for the authors below

General comment

I read your paper with great interest and am convinced it will receive attention when published. On the eve of 2020 and as the FP global community reflects what should be our priorities for the coming decade, it is appropriate to reflect critically on past efforts and implications for the future. This study is indeed very timely.

Thank you for taking the time to review our manuscript and provide the valuable comments. It is really encouraging to hear that as an expert on international family planning, you recognize the timeliness and potential contribution of the study.

As noted at the end of your discussion, despite a leadership role of DFID, and joint efforts to engage USAID and UNFPA in preparing the 2012 London Summit, the FP2020 agenda is closely associated with the Bill & Melinda Gates Foundation and with Melinda herself, a reflection of her determination to empower women and girls to have the number of children they want and when they want them. With the change in administration of the US Government, USAID has been perceived as distancing itself from the FP2020 agenda. This is perhaps signaled by the shaping of a strategy for 2030, and a new indicator and target. With this article, it appears that the authors are reflecting on the dynamics of what are effectively the two powerhouses of the global FP movement: USAID which still is a dominant donor for this work, and the Gates Foundation which has sought to catalyze action and be a strong champion for transforming the approaches of the FP community. By highlighting the divergence in indicators and targets, the authors seem to be making a political comment couched in sophisticated statistics. However, their conclusion is a bit lost. The argument seems to be that

the 75% demand satisfied goals will achieve results faster than numerical targets underpinned by setting goals related to annual percentage increases (I think the original FP2020 metrics group adjusted the annual increase based on recent country performance, which has been somewhat glossed over in this study, some of which is defensible for the clarity of the analysis). However, is that the best course of action? There are important limitations to a complex indicator such as the demand met with modern contraception one (not unlike the unmet need indicator). I hope the authors can sharpen their own position and I look forward to the ensuing debates.

We truly appreciate and completely agree with your comments. You are right about the complex dynamics behind those two initiatives. The 75% satisfied demand target is still being debated, more than five years after it was initially proposed. To our knowledge, however, no alternative targets have been proposed, and attaining 75% satisfied demand by 2030 has been cited as a benchmark in recent studies, such as J New et al. (The Lancet Global Health 2017) and Choi and Fabric (Global Health: Science and Practice 2018). The Sustainable Development Goals 3.7.1 is on “Proportion of women of reproductive age (aged 15-49 years) who have their need for family planning satisfied with modern methods” and the 75% has been proposed as a target level by 2030.

It is beyond the scope of the current study to provide a full account of the history and evolution of the global family planning agenda. Instead, the objective of our statistical exercise is to quantify the concordance of the two initiatives, respectively sponsored by two of the most significant players in international family planning. We have expanded relevant paragraphs in the manuscript, hoping to provide as much background information as feasible.

My chief criticism of the entire approach to this study is that it fails to acknowledge the recent recognition and consensus that mCPR growth is not typically linear in nature. In fact, recent FP2020 metrics experts have determined that country family planning growth patterns follow an S-curve (see FP2020 latest progress report at <http://progress.familyplanning2020.org/content/measurement>). Admittedly, I was not able to find any publication in the scientific, peer-reviewed literature to support these assertions. Yet, all the countries that have experienced bursts of mCPR growth in recent years (first Malawi, Rwanda and Ethiopia and more recently Mozambique, Liberia, Malawi again and Kenya), have been in this middle range of prevalence, in support for the S-curve agreement.

We agree with you on the nonlinearity of mCPR trajectories. And we are aware of the S-curve proposed in the FP2020 progress report, which is a typical curve in logistic growth. However, the assumed S-curve has not been sufficiently supported by the data, which explains the lack of peer-reviewed literature that you mentioned. Validating the S-curve is beyond the scope of the present study. Practically, customizing goals for each country per their current level and recent trend of mCPR may make the targets too technical, and consequently too hard to promote and track. That is why we followed Brown et al. and subsequent studies and simply used an annual growth rate of 1.4 percentage point.

Assuming the S-curve is true, FP2020 countries are at different stages of the curve, some in the convex stage and some in the concave stage. By pooling those countries together, we believe a linear growth may cancel the country-level errors and therefore provide approximately unbiased global estimates.

The architects of the FP2020 goal (the authors cited the Brown et al 2014 paper) constructed the 120 million additional users goal using country projections assumed a linear trend in mCPR growth at the time. I have heard some acknowledge that a major flaw in the calculation of the FP2020 target was the failure to recognize and accommodate for the S-curve pattern. As

a result, the target assigned for growth in India, the country with the largest population in the FP2020 list, was definitely overly ambitious and had a disproportional effect on the overall target. If we could go back in time to reduce the target and align it with a more realistic, yet ambitious target for India, then the numerical goal would perhaps not have been too ambitious, given that the bulk of FP2020's 69 countries are still in the early or middle stages of the S-curve. All this to say that the authors could potentially reference the lessons imparted by the S-curve in their paper, which to my mind, has dramatically transformed target setting for countries developing FP strategies.

We are also aware of the limitations in the methodologies used in the Brown et al.'s original paper. We agree that with the knowledge that we have now, Brown et al. would have developed a different target. Despite later discussion and debates, no substantial updates have been made to the FP2020 global goal and average targets for countries. The target of an annual growth rate of 1.4 percentage point is still being used to monitor progress in FP2020 countries.

The next implication of this is that if, as Cahill et al found, the global mCPR is 45.7%, then the world is approaching the slower part of the S-curve, which may have a dragging effect on achieving either target by 2030. But perhaps this is for another paper.

That is another interesting point. And we agree with you that it is more appropriate for another separate study. The considerable variations in mCPR across countries make it challenging to draw an inference based on a global average.

More detail-oriented, specific comments:

Abstract (and introduction)

The authors characterize family planning as a medical intervention. I suggest you replace term "medical" with "health" as medical implies that a physician has to provide it and some methods, such as fertility awareness methods, have no medical component to them at all, yet arguably impart health benefits.

Thank you for the suggestion. We have changed "medical" to "health".

Nowhere in the abstract is the 2nd global initiative named. When I read it, I recognized it or assumed I recognized it as the Sustainable Development Goals (whose indicator USAID has adopted for its own strategy). I suggest you include greater clarity about who "owns" this target in the text and clarify which initiative you mean in the abstract. As it was only when I got to page 7 or 8 that I realized you were talking about USAID's strategy. Indeed, the target is USAID's as the SDG target is phrased more broadly even if the indicator is common across the two initiatives. Anyway, this vagueness is an issue in my view.

We have revised the abstract to clarify that the 75% satisfied demand initiative was initially proposed and currently owned by USAID. The indicator has been adopted by Sustainable Development Goals (SDG 3.7.1).

Results section of abstract uses the word "latter": Latter is not clear here. The goals are listed in order of FP2020 and SDGs in the methods section. Why not be explicit and say FP2020?

We have revised the sentence accordingly.

Introduction

The summation of the Cahill et al conclusion that the FP2020 goals were overambitious somewhat misrepresents the nuance of what that paper seemed to convey. I believe they allude to the diversity of countries and the demographic pull of large countries on the targets.

Cahill et al. (2018) has several findings. We are only citing one of them that is most relevant to our exercise. As you mentioned, Cahill et al. also shows the diversity in progress at the country level. We have revised the sentence to avoid confusion.

Data

You explain that the analysis includes data from 67 countries (page 6, line 49, 1st instance – appears again in Methods section (page 9, line 27), yet in the Results (page 11, line 8, 1st instance), you refer to 66 countries. What happened to the missing country?

Sorry for the typo. Our analysis is based on 67 countries. The manuscript has been updated.

Results

Incomplete sentence on page 11, lines 42-45.

Sorry for that. The sentence has been completed.

Limitations

The strong fit of the model may imply that I am wrong to make this comment, however, I wonder if the authors can comment on the linearity of the modeling assumptions as opposed to the typical patterns in mCPR growth.

The observed relationship between mCPR and the % satisfied demand varies greatly across countries and over time. Empirically, an accurate description of their relationship requires an incredibly complex formation (e.g., time-varying country-specific random intercepts and random slopes), which may not be identifiable using the observed data. Given our focus on global instead of country-specific results, we can afford to lose certain country-specific precision as long as the country-specific deviations are approximately random. Our quadratic formulation achieved this goal, which is evident from the strong model fit.

Discussion

Page 13, second paragraph. The authors write: “On the other hand, achieving a 1.5% annual increase in all-woman mCPR will enable less than half of the 41 pledging countries to attain the goal of 75% satisfied demand by 2030.” I am troubled by the over-simplification of the target-setting that this sentence implies. I don’t think that this how Brown et al described the target setting for the FP2020 target or how countries go about setting targets. Authors should acknowledge that the 1.5% measure is one that they adopted as a convenience for their study. And that it doesn’t pass the test for either global or country targets without understanding the stage a country is in and recent patterns of growth, the demographic or youth bulge that it must also deal with.

The original article by Brown et al. found that “0.7 percent per year was the overall mCPR annual growth rate across all 69 countries before 2012”, and then proposed that “an aspirational yet achievable goal would be to realize an approximate doubling of the average annual MCPR growth from 0.7 to 1.4 percentage points by 2020 across all 69 countries.” The value 1.4 has been frequently used in discussions about FP2020.

Brown et al. emphasized that “it was therefore critical to formulate the overall goal in a way that would not be construed as a series of country-specific targets”. In our statistical exercise we simply assume that the aspirational goal of an annual increase of 1.4 percentage point is achieved in all the FP2020 countries.

We have clarified the rationale for the 1.4 measure and provided more background information, hoping to avoid any misunderstanding or confusion.

Conclusion statement in abstract. I missed this statement in the discussion or a strong link back to the title of the paper, so we are left with the somewhat weak “conclusion” in the abstract. Are the authors arguing for further convergence in the post-2020 period? The statement implies that the community should mobilize around both metrics. Given the title of the paper uses the term “shared goals”, I was hoping the authors would take more of a position given the critical timing of this paper.

Our objective is to strengthen the coordination of the two initiatives by illustrating the concordance of their shared goals. We have expanded the Discussion session following your suggestions.

Tables and figures

Table 1 and 3: I suggest replacing the term “pledging” with the more common language of “commitment-making” countries

Thank you for the suggestion. We have changed “pledging” to “commitment-making.”

Table 3: The title of this table doesn’t explain well what data are being presented here. If I understood correctly, the 2020 and 2030 data are estimates assuming either achievement of 75% demand satisfied by 2030. I inferred this from the text. However, data in tables should stand alone and be interpretable without reading the text, thus the title should be made clearer as to the analysis performed to arrive at these estimates.

You are right that Table 3 shows the results assuming the achievement of 75% satisfied demand by 2030. We have revised the caption of the table.

Table 4. Similar comment as Table 3

The caption of the Table 4 has been updated.

Figure 2: I suggest you add a horizontal line at 0, or move the vertical legend so that 0 is at the corner, otherwise it visually looks like the non-commitment-making countries are contributing more than the space (difference) between the commitment-makers and the total.

Thank you for the suggestion. We have added a horizontal line at 0.

Additional comments extracted from the reviewer’s annotations on the pdf document

linearity vs s-curve pattern?

We have added the potential model misspecification as another limitation of our study.

but i think even the FP Metrics group would acknowledge flaw in their reasoning. And the demand indicator is as complex than the unmet need indicator that was rejected. My personal experience is that it is difficult to incorporate a demand met with modern contraceptive goal in a country strategy, whereas a numerical target for CPR is a common attribute of CIPs.

Thank you for sharing your insights. The Choice of Metric section in Brown et al. (2014) detailed their deliberations on indicator selection. Despite the limitations of met (or unmet) need that you and Brown et al. mentioned, USAID is still leading the global effort on achieving 75% satisfied demand. And the indicator has been adopted as an SDG indicator for family planning. Since FP2020 continues using mCPR or users, a statistical exercise like ours is more warranted to explore the correlation of mCPR (or users) and met (or unmet) need.

Responses to Reviewer 3's Comments

Reviewer Name: Asad Khan

Institution and Country: The University of Queensland

Australia

Please state any competing interests or state 'None declared': None to declared

Please leave your comments for the authors below

Insufficient information about statistical methods and data.

Thank you for the comments. We have added more details about the methods and data.

Responses to Reviewer 4's Comments

Reviewer Name: Tesfalide Tekelab

Institution and Country: University of Newcastle, Australia and Wollega University, Ethiopia

Please state any competing interests or state 'None declared': None declared

Please leave your comments for the authors below

I found the manuscript interesting. It is important topic. I have some minor comments.

1. The conclusion is not written clearly. I have seen only the discussion part. Write the conclusion inline with your finding

We are really glad that you found our study interesting. And thank you for the suggestion. We have expanded the Discussion section.

2. Line 42-45 – Needs revision. The sentence is not complete. “Another 9 countries (Ethiopia, Laos, Madagascar, Nepal, Philippines, Rwanda, Solomon Islands, Tanzania, and Zambia)”

Sorry for the mistake. The sentence has been completed.

3. Line 52- Change the word “discus” to discuss.

Done.

Responses to Reviewer 5's Comments

Reviewer Name: Andrew Hinde

Institution and Country: University of Southampton, United Kingdom

Please state any competing interests or state 'None declared': None declared

Please leave your comments for the authors below

This paper examines the concordance between two family planning initiatives: (1) the addition of 120 million contraceptive users by 2020, and (2) satisfying 75 per cent of the demand for modern contraception among married or in-union women aged 15-49 years by 2030. It seems an interesting exercise to assess the implied consistency between the objectives of these initiatives, for there should be some relationship between them. Indeed, one of my suggestions is that you explore the theoretical relationship between them before embarking on your empirical analysis.

Thank for the suggestion. We have expanded the discussion on the theoretical relationship between the two indicators.

Unfortunately, the paper as it stands is hard to follow and has some serious weaknesses. In my opinion it requires substantial revision before it could be published. I have five general points to make and a number of specific issues with individual passages or sections.

General points

1. My first point is very basic. Do the 120 million contraceptive users to be added include males as well as females? If female, are they supposed to be married or in-union, or do any women who are persuaded to start using contraception count towards the target? (Actually, I know that the 120 million only includes women, but you might state this, for if they do include males much of the basis of your paper is undermined.)

You are right that the 120 million goal counts all women, regardless of their marital or cohabitation status. And males are excluded. We have clarified this in the manuscript.

2. Before embarking on your empirical analysis, it might be worth a brief theoretical exploration of the relationship between the various quantities you describe in the paper, taking the case of a single country. This will help you define quantities clearly, and establish in the minds of readers the difference between them. For example, if the number of (married or in-union?) women aged 15-49 years using modern contraception is C, the number of women who are not using but who do not wish to become pregnant at the present time is U, and the number who are trying to become pregnant or who do not wish to use contraception for other reasons is N, then the proportion of demand satisfied is $C/(C + U)$. The prevalence rate is $C/(C + U + N)$. Of course, you have data for many countries, and hence you use statistical methods to establish the 'average' relationship across these countries. But the results should still not be too different from the theoretical relationships.

Thank you for the great suggestion. We have added an example to illustrate the quantities and their relationship.

3. The theoretical excursion mentioned above would help you sort out one of the main weaknesses of the paper, which is the ambiguous or unclear definition of quantities. On p. 5,

II. 17-21, for example, there is confusion between 'all women' and 'women in a sexual union'. The paper refers to the percentage of demand satisfied as the 'proportion of all women who use modern contraception divided by the total demand for family planning' (II. 18-19) but then defines the total demand for family planning as equal to the sum of the 'percentage of married or in-union women aged 15-49 who are using any contraception' (II. 19-20) and the 'percentage of all women with unmet need' (I. 20). I was lost at this point. Can you define quantities and express what you mean precisely? The quantities should be defined in numbers of women, and you should clearly state in each case the age range you are considering, and whether you are restricting attention to married or in-union women.

Thank you for the suggestions. We have added clear definitions of the indicators.

4. As I understand your method, you use past data to estimate some coefficients. You use the latest contraceptive prevalence rate as a baseline and predict the contraceptive prevalence in 2030 using your estimated coefficients. This involves a big assumption that your coefficients will not change between now and 2030. On p. 8, II. 11-12 you suggest that you will be predicting out of the range of your data (you will be using your model to make out of sample predictions). I think you need to do more to persuade me (and the reader) that you have accounted for this requirement when estimating the model. How do you assess the predictive performance of the model? What methods did you use? Did you use cross-validation, for example leave one out validation? On p. 12, II. 11-12 you say that you are 'mainly interested in the predictive performance of the model measured by the adjusted R-squared'. Predictive performance would be better assessed using cross-validation and the mean square error or the Akaike Information Criterion than by just using the R-squared (even though it is adjusted) on the model fit to past data.

Our study indeed requires the assumption that the relationship between mCPR and satisfied demand estimated from historical data applies to the future. And we consider it a plausible assumption. The projection period (2018-2030, or 13 years) is relatively short compared to the historical period (1986-2016, or 31 years). There are unlikely any structural changes that will fundamentally alter the relationship between mCPR and satisfied demand.

Regarding model validation, we have tried not only the leave-one-out cross validation but also the forward projection. For example, we used 1986-2010 data to estimate the model and project for 2011-2016. The forward projections verified the strong predictive performance of the models. Those projection results also support the assumption above that the relationship between mCPR and satisfied demand is relatively stable over time. The results from model validations are omitted in the manuscript because we believe the close-to-one adjusted R-squared demonstrated the same information, though from a different perspective.

5. The paper's conclusion, that 'a broad coalition needs to be formed to accomplish both initiatives' (p. 3, II. 3-4) is hardly earth-shattering. I could have written that without needing all your analysis. What is new in your paper that needs emphasizing? My take on your results is that the 75 per cent of demand satisfied goal is considerably more ambitious than just adding 120 million new users. This is worth emphasizing.

We agree that the need to form a broad coalition is largely a political or policy issue that does not need any complicated justification. The contribution of our statistical exercise is to quantify implications of the two initiatives on each other. We have revised the title, expanded relevant sections, adding a discussion on the result that the 75% satisfied demand goal is overall more ambitious the FP2020 in many countries.

Specific points

p. 4, I. 20 Insert 'more' after 'planning is'.

Done.

p. 5, I. 11 'priorities are' should be 'priorities is'.

Corrected.

p. 5, I. 21 'any contraception' should, I think, be 'any modern contraception'.

It should be "any contraception" because women using traditional contraception are also included in the total demand.

p. 5, I. 21 How do you measure 'unmet need'? This is an important issue for your paper, so could you explain how it is defined and calculated?

Unmet need is defined as the proportion of women who want to stop or delay childbearing but are not using any method of contraception. We have added this information to the manuscript.

p. 6, II. 13-4 What are these 466 surveys? Are they Demographic and Health Surveys (DHSs), Multiple Indicator Cluster Surveys (MICS), or other surveys? How many of them are DHSs or MICS?

This dataset of 466 surveys is assembled by Population Division at the United Nations (see reference 12). Much of the information was obtained from multi-country survey programs (e.g., DHS; MICS) and additional information was provided by other international survey programs and national surveys.

p. 7, I. 6 'assuming a 1.5% annual increase'. Do you mean 1.5% or 1.5 percentage points? See also p. 10, I. 1.

We meant an annual increase of 1.5 percentage points. As discussed elsewhere in the responses to reviewers, we have changed from 1.5 to 1.4. We have revised the manuscript to clarify.

p. 7, II. 18-19 It seems unnecessarily conservative to assume that the contraceptive prevalence rate and the percentage of demand satisfied will remain constant until 2030 for these well provided countries.

We agree that this assumption sounds over-conservative. We made this assumption for two reasons. First, a country usually "graduates" from the initiative once they reach the target, and therefore no longer requires international support. And the objective of our exercise is to assess the concordance of international initiatives that aim to reach the set targets. Second, it is hard to assign a trajectory once a country reaches the target. A plateauing curve is commonly observed among the world's best performers, and therefore a stable trend seems to make sense after reaching the target.

p. 8, II. 2-3 'For the other 36 countries, the percentage of FP demand satisfied with modern methods will reach 75% in 2030'. How do you know? Is this an assumption, and how does this relate to the 1.5% (or 1.5 percentage point) increase mentioned on p. 7, I. 6.

Satisfying 75% of the FP demand by 2030 is the target of the 75% satisfied demand initiative. And the 1.4% annual increase is the proposed goal of FP2020. We are assessing the two initiatives' concordance by simulating the implication of achieving one initiative on the other.

p. 8, I. 8 'The mode is chosen from several options' should be 'The model is chosen from several options'.

Sorry for the typo. It has been corrected.

p. 8, II. 15-18 I presume that the reason you have to use two steps in this stage is that you do not have DHS data for all your 36 countries. Or, more accurately, 204 out of your 466 surveys are not DHSs based on samples of all women. If you used only the 262 DHSs based on samples of all women, you could do all this in one step, for the DHS data would allow you to compute the contraceptive prevalence for married and in-union women as well as all women. Can you explain why it is better to use the extra 204 surveys even though it makes the whole process more complicated and possibly less accurate?

As you mentioned, our two-step approach allows us to use the full dataset (i.e. 466 surveys), which likely improves model performance than only using a subset (i.e. 262 surveys). Our approach may result in more accurately predicted married mCPR from % satisfied demand. The complexity due to this two-step approach is necessary because our findings critically depend on the predictive accuracy of the models.

pp. 8-9 The symbols in the equations do not always mean the same thing. In equation (1) (p. 8, I. 5), y_{it} refers to the contraceptive prevalence rate (CPR), and x_{it} is the percentage of demand satisfied. However in equation (2) x_i is the married and in-union CPR and y_i is the all-women contraceptive prevalence rate. This is confusing for the reader. A symbol should mean the same thing throughout your paper. On p. 10, II. 3-4 you say you are 'reversing the meaning of y_{it} and x_{it} '. Do not do this! Keep the meaning of the symbols the same and change the equation.

We thank you for the suggestion to standardize the notations, and we have revised the formulas accordingly.

p. 9, II. 12-14 Why did you exclude the five countries who had reached the 75 per cent demand satisfied goal? They contribute 100 per cent achievement of the goal.

From the perspective of the 75% satisfied initiative, those countries will not be the focus of initiative activities. Therefore, they are excluded from the current exercise. Clearly the percentage of countries reaching the 75% target will be higher if those countries are accounted for. We have revised the manuscript to add them back the discussions.

p. 10, I. 7 Change 'involv'd' to 'involved'.

Corrected.

p. 10, II. 12-13 'less than 2% of the variations in all-woman mCPR cannot be

explained by the model'. This is very high. It suggests that contraceptive prevalence and the percentage of demand satisfied are very closely related. Does this not immediately suggest that there is a high degree of concordance between the two?

The main reason for the high adjusted R-squared is actually due to accounting for the country dummy (or country-level fixed effects). In statistical terms, the number of parameters is only one for the variable of % satisfied demand and 66 for the country dummy variable (one fewer than the number of countries because country dummies require a base country). Despite the theoretically strong correlation between mCPR and % satisfied demand, their bivariate correlation has been found weak in empirical studies.

p. 11, ll. 5-7 'Following the trajectory of increasing mCPR and % satisfied demand, the goal of adding 120 million modern contraceptive users will be achieved in early 2023'. I

do not understand this sentence.

We mean that if the countries could continue the growth rate of mCPR implied by the 75% satisfied demand initiative, 120 million modern contraceptive users will be added in the 67 countries by early 2023. We have revised the sentence to clarify.

p. 11, ll. 15-16 'Another 9 countries ...' will do what?

Sorry for the mistake. We have completed the sentence.

p. 18, Table 3. There are several issues with this table. (1) How can Chad have a contraceptive prevalence rate that is negative?

Our model does not include any restriction on the outcome value. Therefore, it is almost inevitable to have out-of-range predicted values. Since the predicted mCPR for Chad is only -0.4%, we believe it makes sense to round it 0.

(2) The figures in for Bangladesh seem to be inconsistent: $17,800 + 2,366 \neq 20,200$ and $17,800 + 4,369 \neq 22,200$.

Thank you for checking our results so carefully. The inconsistencies are due to rounding errors. Exact numbers are used in the model while results are presented in thousands. For Bangladesh, the exact numbers are $17790088+2366236=20156324$; $17790088+4369472=22159560$. We have added a note under the table to avoid confusion.

(3) Have you considered comparing your married/in-union contraceptive prevalence rates estimated for 2020 with those for the latest DHSs for those countries where recent DHSs have been held? This is not difficult or time-consuming to do. Actually it is so quick that I did it in 6 minutes for the first 10 countries on your list using the Statcompiler on the DHS web site. Generally, for those countries with a DHS since 2015 the latest modern contraceptive prevalence rate is consistent with your 2020 predictions, though there are exceptions. For example, Ethiopia's modern contraceptive prevalence was 35 per cent in

2016, suggesting that your prediction of 32.6 per cent by 2020 is too low. You could do this kind of comparison for the predictions for 2020 in Table 3, p. 18 and Table 4, p. 21, to see how far adrift each country is likely to be. Of course, you cannot do it for all countries, but you can do it for several countries with DHSs in 2016, 2017 or 2018.

Thank you for the great suggestion. The slight underestimation by our model is expected given the intensified focus on the FP2020 priority countries, including Ethiopia. This does not affect our conclusion because our exercise is mainly a numerical simulation based on the proposed goals of the two initiatives. We have added this interesting point to the manuscript.

VERSION 2 – REVIEW

REVIEWER	Kirsten Black university of Sydney
REVIEW RETURNED	01-Sep-2019

GENERAL COMMENTS	Thank you for improving the manuscript. On the whole this version is much clearer. There remains some misalignment between aims, results and conclusions and there are still a number of syntax errors particularly in regard to use of american english. Particular comments are Abstract last line of the methods "confirm models' performance". Is this one model or multiple? Results: The results are much clearer when described on page 14 (para 2 of the discussion). Conclusion: The conclusion still does not really answer with the objective. The conclusion could read as in the discussion "The results show that the two initiatives move towards the same goal of promoting access to FP and overall both are ambitious. Introduction The introduction read well Methods: improved Discussion: Overall the discussion has improved and a clear outline of the policy implications enhance the paper. I would recommend some further changes however. The first paragraph is not clear. The sentence for example that re-iterates the objectives "We estimate and discuss the implication.." is not required. Please consider stating the findings (currently in a well written second paragraph) in the first paragraph as this will build the case for the contribution. The conclusion at the end of the discussion differs from the conclusion in the abstract.
---

REVIEWER	Andrew Hinde University of Southampton United Kingdom
REVIEW RETURNED	04-Oct-2019

GENERAL COMMENTS	Thank you for sending such a comprehensive and detailed letter with the revised version. This version of the paper is dramatically improved compared with the previous version and I recommend it be accepted. I have just a couple of suggestions: on p. 2, l. 6 insert 'female' before 'modern contraceptive users'; also
---

	on p. 10, l. 18 p. 3, l. 13 define 'mCPR', as some readers will not have met this measure before
--	---

VERSION 2 – AUTHOR RESPONSE

Responses to Reviewer 1's Comments

Reviewer Name: Kirsten Black

Institution and Country: University of Sydney, Australia

Please state any competing interests or state 'None declared': None declared

Thank you for improving the manuscript. On the whole this version is much clearer. There remains some misalignment between aims, results and conclusions and there are still a number of syntax errors particularly in regard to use of american english. Particular comments are

We are very glad to learn that our revisions addressed your previous concerns. We have thoroughly proofread the manuscript again during this round of revision.

Abstract

last line of the methods "confirm models' performance". Is this one model or multiple?

As described in the Methods section and illustrated in Figure 1, multiple statistical models were used in the study. We checked and confirmed the predictive performance for each of them.

Results: The results are much clearer when described on page 14 (para 2 of the discussion).

We agree that some information in the second paragraph of the Discussion section should be presented earlier. And we have added that information to the Results section.

Conclusion: The conclusion still does not really answer with the objective. The conclusion could read as in the discussion "The results show that the two initiatives move towards the same goal of promoting access to FP and overall both are ambitious.

Thanks for the suggestion. We have slightly expanded the Conclusions section accordingly.

Introduction

The introduction read well

Methods: improved

Discussion: Overall the discussion has improved and a clear outline of the policy implications enhance the paper. I would recommend some further changes however. The first paragraph is not clear. The sentence for example that re-iterates the objectives "We estimate and discuss the implication.." is not required. Please consider stating the findings (currently in a well written second paragraph) in the first paragraph as this will build the case for the contribution.

Thanks for the suggestions. We have removed the unnecessary sentence and re-arranged a few sentences to smoothen the flow.

The conclusion at the end of the discussion differs from the conclusion in the abstract.

We have revised the relevant sentences to harmonize the message.

Responses to Reviewer 5's Comments

Reviewer Name: Andrew Hinde

Institution and Country: University of Southampton, United Kingdom

Please state any competing interests or state 'None declared': None declared

Thank you for sending such a comprehensive and detailed letter with the revised version. This version of the paper is dramatically improved compared with the previous version and I recommend it be accepted. I have just a couple of suggestions:

We are very glad to hear you find our revisions satisfactory.

on p. 2, l. 6 insert 'female' before 'modern contraceptive users'; also on p. 10, l. 18

Done.

p. 3, l. 13 define 'mCPR', as some readers will not have met this measure before

Done.

VERSION 3 – REVIEW

REVIEWER	Kirsten Black University of Sydney
REVIEW RETURNED	15-Oct-2019

GENERAL COMMENTS	Thank you. There has been much improvement but please correct the syntax issues. The spelling has not been changed British english. Please rephrase the following "The results from the statistical exercise demonstrate that the two global initiatives move toward the same goal of promoting access to FP and overall both are ambitious" replace with "The results from this study...." or "The results from the statistical modelling.."
---